# A Lightweight Obstructive Sleep Apnea Detection Method with Millimeter-Wave Radar and Oximeter

Long Pu
Department of Electronic Engineering
Tsinghua University,
Beijing, China
pl23@mails.tsinghua.edu.cn

Xueqian Wang
Department of Electronic Engineering,
and also with the State Key Laboratory
of Space Network and Communications
Tsinghua University
Beijing, China
wangxueqian@mail.tsinghua.edu.cn

Zetao Wang
Beijing Tsingray Technology Co., Ltd.
Beijing, China
wangzetao@qingleitech.com

Zhaoxi Chen
Beijing Tsingray Technology Co., Ltd.
Beijing, China
chenzhaoxi@qingleitech.com

Jian Guan
Shanghai Sixth People's Hospital
Affiliated to Shanghai Jiao Tong
University School of Medicine
Shanghai, China
guanjian0606@sina.com

Gang Li
Department of Electronic Engineering,
and also with the State Key Laboratory
of Space Network and Communications
Tsinghua University
Beijing, China
gangli@mail.tsinghua.edu.cn

*Abstract*—**Obstructive Sleep Apnea (OSA) is a prevalent disorder characterized by intermittent cessation of breathing during sleep. The established gold standard for OSA diagnosis, Polysomnography (PSG), is uncomfortable for patients. This paper proposes a user-friendly and fine-grained method for detecting OSA events with millimeter-wave radar and oximeter. To adequately fuse the two sensors, we introduce the multi-scale feature extraction strategy and neighboring short-term feature enhancement strategy (Ms&Ne). Key features are extracted at long, medium, and short scales, capturing both long-term characteristics and detailed variations of the signals, effectively addresses the signal misalignment issue due to oxygen desaturation delay. Short-scale features are further incorporated to enhance short-term variation detection. The eXtreme Gradient Boosting (XGBoost) is utilized for tree-based feature interactions. Clinical trials involving 121 patients at Shanghai Sixth People's Hospital demonstrate that our method achieves a highest F1-score of 0.7713 for OSA detection at the second-by-second level.**

*Keywords—obstructive sleep apnea, millimeter-wave radar, oxygen saturation, feature fusion*

## I. INTRODUCTION

Obstructive sleep apnea (OSA) is a common sleep disease that causes daytime sleepiness, metabolic disorders, hypertension, etc., and is closely related to the incidence of cardiovascular diseases [1]. Currently, the gold standard for diagnosing OSA is Polysomnography (PSG) [2]. It is difficult for PSG to popularize due to its over-reliance on specialized equipment and personnel [3], as well as the discomfort of the diagnostic process [4], [5].

Driven by the demand for diagnostic comfort, the contactless technique for detecting OSA, developed based on infrared video [6], [7], piezoelectric sensors [8], [9], acoustic sensors [10], [11], and radar, has gained increasing preference. Compared with other sensors, millimeter-wave radar exhibits superior anti-interference capabilities and can operate stably in low-light conditions and noisy backgrounds [12], while protecting the privacy of patients. Millimeter-wave radar captures tiny thoracoabdominal movements of patients during sleep, and detect OSA through either threshold-based methods [13], [14] [15] or feature based methods [16], [17]. Zhuang et al. [16] extracted empirical features from respiratory signals obtained from FMCW radar and trained a random forest (RF) model to classify signal segments into apnea and non-apnea

categories. Choi et al. [18] employed a recurrent neural network (RNN) to extract features from the respiratory signals automatically, reducing the need of prior knowledge. However, radar signals are susceptible to body movements of patients, and it is difficult to detect some special apneas that airflow ceases while respiratory effort remains [19] or subtle apnea types like hypopnea. The fusion of radar and oxygen saturation (SpO2) can largely compensate for the shortcomings of radars [20]. SpO2, as an internal physiological parameters [21], helps the radar recall some subtle sleep apnea events. Meanwhile, SpO2 collection may be affected by poor peripheral arterial blood flow, vasoconstriction and hypotension [22], as the respiratory signals extracted from radar compensate for these limitations. Toften et al. [23] are the first to integrate radar and SpO2 signals, who conducted a comparative experiment on 14 patients and demonstrated that the fused signals improved the classification accuracy of apnea event types. However, the authors did not provide a detailed description of the methods used in [23]. Ma et al. [19] employed the combination of CNN and long short-term memory (LSTM) to extract features from radar and SpO2 signals, and performed fusion at both the feature level and decision level respectively, achieving an F1-score of 0.76 for classifying apnea and non-apnea segments. However, the method in [19] is based on segment analysis, where continuous radar and SpO2 signals are divided into 30-second segments for classification. Each segment is analyzed in isolation with a limited receptive field of 30 seconds, lacking contextual information between neighboring segments, which is crucial in OSA detecting. The limited receptive field causes the key features of SpO2 and radar to be fragmented into different segments due to the delay of oxygen desaturation, leading to fusion failure. Wang et al. [20] utilized radar spectrograms to detect OSA at the event level, ensuring event integrity, and ultimately fused SpO2 information at the decision level. However, the decision-level fusion method was unable to integrate the features of radar and SpO2 signals adequately, and more fine-grained comparisons beyond event-level results are required in [20].

In this work, we adopt the multi-scale feature extraction and neighboring short-term feature enhancement (Ms&Ne) strategy and second-by-second analysis to address the issues of SpO2 latency-induced feature misalignment, and coarse granularity limitations inherent in prior study.

## II. MATERIALS AND METHODS

### A. Data collection

Participants in this study were monitored by oximeter (from PSG) and millimeter-wave radar simultaneously. Fig. 1a illustrates the placement of the experimental equipment. PSG sleep monitoring was performed on Philips Alice 6. The millimeter-wave radar used in this study is QSA600, developed by Beijing Tsingray Technology Co., Ltd., see Fig. 1b. QSA600 is a sleeping breathing monitoring system developed based on Infineon BGT60TR13C, an FMCW radar, see Fig. 1c. FMCW radar offers high resolution and low power consumption, making it suitable for monitoring vital signals [14]. The key parameters of BGT60TR13C are shown in TABLE I. QSA600 is installed 1 meter above the head of the bed, allowing the radar beam to effectively cover the thorax and abdomen of subjects, see Fig. 1a.

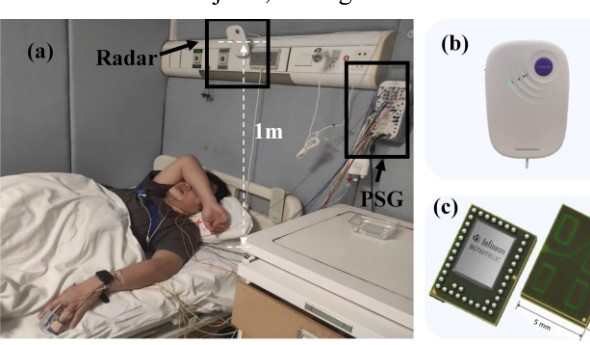

Fig. 1. Schematic diagram of experimental equipment and scene. (a) the positions of the patients, PSG equipment and millimeter-wave radar, (b) QSA600, (c) Infineon BGT60TR13C.

TABLE I
KEY PARAMETERS OF BGT60TR13C

| Parameters | Values |
| --- | --- |
| Start frequency | 58 GHz |
| Bandwidth | 3.75 GHz |
| Frame rate | 250 Hz |
| Slope | 15 MHz/us |
| Sampling frequency | 1 MHz |
| Chirp period | 750 us |
| Chirps per frame | 1 |
| Range resolution | 0.04m |

Clinical data were collected at Shanghai Sixth People's Hospital Affiliated to Shanghai Jiao Tong University School of Medicine. Of 168 initially collected recordings, 121 qualified for analysis after excluding: 23 due to sensor detachment, 5 for signal interruption, 4 with SpO2 artifacts, and 15 for other technical failures. Among the 121 patients (mean age $37.5 \pm 13.1$ years, range from 19 to 76; mean BMI $24.5 \pm 3.6$; 67.8% male), 23 were healthy (AHI < 5), 32 had mild OSA ($5 \leqslant$ AHI < 15), 17 had moderate OSA ($15 \leqslant$ AHI < 30), and 49 had severe OSA (AHI $\geqslant$ 30). The overall mean AHI was $30.0 \pm 27.5$. Sleep monitoring revealed a mean recorded time of 9.7 hours per night and a mean sleep time of 7.1 hours, with apnea time (positive sample) accounting for 16.1% of total recorded time. All valid PSG recordings were annotated with a consensus result from two certified sleep technicians, following American Academy of Sleep Medicine (AASM) v2.6 recommendations [24]. This study was conducted in accordance with the principles of the Declaration of Helsinki and the Good Clinical Practice for Medical Device Trials. It received ethical approval from the Ethics Committee of Shanghai Sixth People's Hospital Affiliated to Shanghai Jiao Tong University School of Medicine (2023-030-[1]), and has been registered with the U.S. Clinical Trials Registry (NCT06038006).

### B. Extracting respiratory signal from radar

We preprocess the raw radar echo data into respiratory signals [16], as shown in Fig. 2. The intermediate frequency (IF) signal is converted from analog to digital, at a sampling rate of 50Hz. Then, we perform range fast Fourier transform (FFT) to obtain movement information of patients over time in different range bins, contained in a range-time domain matrix $\mathbf{X} \in \mathbb{C}^{64 \times N_0}$, where 64 is the number of range bins and $N_0$ is the number of sampling points. And we perform the moving target indicator (MTI) process to eliminate background clutter. We extract the background clutter $x_{clu,(k,n)}$ using the time accumulation strategy, which can be expressed as a recursion,

$$x_{clu,(k,n)} = \alpha \cdot x_{(k,n)} + (1-\alpha) \cdot x_{clu,(k,n-1)} \quad (1)$$

where $x_{(k,n)}$ denotes the element in $\mathbf{X}$, and $x_{clu,(k,n)}$ represents the clutter (assuming $x_{clu,(k,0)} = 0$), $k$ is the index of range bin for $k = 1,...,64$, and $n$ represents the index of sampling points. We set $\alpha$ to 0.01, helping the accumulation pay more attention to static clutter. Accumulation operation in (1) enhances the static target signals, and finally we obtain the static clutter $\mathbf{X}_{clu}$. We remove $\mathbf{X}_{clu}$ from $\mathbf{X}$, then we can get the moving target signal $\mathbf{X}_{MTI} \in \mathbb{C}^{64 \times N_0}$,

$$\mathbf{X}_{MTI} = \mathbf{X} - \mathbf{X}_{clu} \quad (2)$$

After MTI, we resample the signal to 10 Hz and calculate the phase difference $\Delta\varphi$ along the time dimension,

$$\Delta\varphi_{(k,n)} = angle(\frac{x_{MTI,(k,n)} \times x_{MTI,(k,n+1)}^*}{\sqrt{\left|x_{MTI,(k,n)}\right|\left|x_{MTI,(k,n+1)}^*\right|}}) \quad (3)$$

where $x_{MTI,(k,n)}$ denotes the element in $\mathbf{X}_{MTI}$, $^*$ represents conjugate, $|.|$ means calculating the magnitude, and $angle(.)$ is used to calculate the phase angle of the complex number. Then we obtain the final phase difference matrix $\Delta\mathbf{\Phi} \in \mathbb{R}^{64 \times N}$,

$$\Delta\mathbf{\Phi}[k,n] = \Delta\varphi_{(k,n)} \quad (4)$$

Patients are not in a fixed range bin during sleep. Therefore, we selected multiple range bins between 0.5m~1.5m, and performed a weighted summation based on energy of range bins. The weighting matrix $\mathbf{W}$ and weighted phase difference matrix $\Delta\mathbf{\Phi}_W$ can be represented as

$$\mathbf{W}[k,n] = \frac{\left|\Delta\varphi_{(k,n)}\right|^2}{\sum_i \left|\Delta\varphi_{(i,n)}\right|^2} \quad (5)$$

$$\Delta\mathbf{\Phi}_W = \Delta\mathbf{\Phi} \odot \mathbf{W} \quad (6)$$

where $\odot$ denotes Hadamard product. Then, $\Delta\mathbf{\Phi}_W$ is summed along the range bin, yielding weighted phase vector $\Delta\mathbf{\varphi}_W \in \mathbb{R}^N$,

$$\Delta\boldsymbol{\varphi}_W = \sum_j \Delta\boldsymbol{\Phi}_W(:,j) \qquad (7)$$

The energy-based weighted summation effectively resisting the maladaptation of the fixed range bin method caused by different sleeping positions of different subjects.

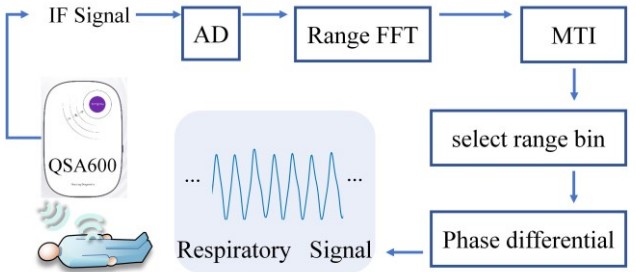

Fig. 2. The preprocessing method of radar echo data: extracting respiratory signals.

Subsequently, we obtain the thoracoabdominal motion[25] $\Delta\mathbf{d} \in \mathbb{R}^N$ and the respiratory signal $\mathbf{r}_{es} \in \mathbb{R}^N$,

$$\Delta\mathbf{d}[n] = \frac{\lambda}{4\pi}\Delta\boldsymbol{\varphi}_W[n] \qquad (8)$$

$$\mathbf{r}_{es} = \Delta\mathbf{d}[n] \qquad (9)$$

where $\lambda$ is the wavelength of the radar beam.

### C. Feature engineering

#### 1) Feature selection

Respiratory signals from radar and SpO2 signals from PSG are both time sequences, containing rich time and frequency features. In this study, we first extract 473 relevant features for respiratory signals and SpO2 signals from a machine learning (ML) feature library called tsfresh [16] respectively. Tsfersh contains rich time series features from both time and frequency domain. We selected features with the Pearson correlation coefficients feature screening method in [26], and finally retained 40 features for respiratory signals and SpO2 signals respectively. All 80 selected features and their descriptions are listed in TABLE II.

#### 2) Multi-scale feature extraction (Ms)

According to the guidelines established by AASM, technicians need to take into account contextual information when annotating an apnea event. It is necessary to widen the receptive field when detecting apnea events. In this study, we extract features at three scales using different windows that are comprehensive to encompass most of the useful contextual information around each second, thereby expanding the receptive field of the model.

As shown in Fig. 3, windows at three different scales, are utilized to extract features. Compared to a single window [19], multi-scale windows allow for the capture of short-term, medium-term, and long-term characteristics in the respiratory and SpO2 signals. And the delayed SpO2 features will not be split into the next segment. The sliding step for multi-scale windows is set to 1 second. After multi-scale feature extraction, we can obtain the feature matrix $\mathbf{H}_{01} \in \mathbb{R}^{N \times 120}$ for respiratory signals and $\mathbf{H}_{02} \in \mathbb{R}^{N \times 120}$ for SpO2 signals.

#### 3) Neighboring short-term feature enhancement (Ne)

Fig. 4 shows the process of neighboring short-term feature enhancement (Ne). Short-term features can directly reflect the intrinsic characteristics of current second. Therefore, we concatenate the short-term features of each second with those of the seconds located 15 seconds apart from it, further complementing the characteristics of the neighboring seconds to better capture the variation within the signals, as shown in Fig. 4b.

TABLE II
Selected features of respiratory/SpO2 Signals

| Features of respiratory signal | | Features of SpO2 signal | |
|---|---|---|---|
| 1 | Kurtosis | 1 | absolute sum of changes |
| 2 | binned entropy | 2~23 | change quantiles |
| 3~10 | linear trend | 24 | sum of reoccurring values |
| 11~12 | change quantiles | 25~26 | reversal asymmetry statistic |
| 13 | autocorrelation | 27 | mean abs change |
| 14~19 | permutation entropy | 28~33 | FFT coefficient |
| 20 | absolute maximum | 34~37 | FFT aggregated |
| 21 | minimum | 38 | std |
| 22 | energy ratio by chunks | 39 | variation coefficient |
| 23 | spkt welch density | 40 | variance |
| 24~25 | FFT coefficient | | |
| 26~28 | peak number | | |
| 29 | Fourier entropy | | |
| 30~35 | quantile index | | |
| 36 | large std | | |
| 37 | mean of absolute max | | |
| 38~40 | ration beyond r | | |

Note: some features possess more than one, due to their adjustable parameters.

The respiratory signals from the radar fluctuate in real time when an apnea event occurs, whereas the SpO2 signals often exhibit a delay of approximately 15 seconds [27]. Therefore, we designed mode 2 for SpO2 signals by concatenating the short-term features of each second with those from the 15th and 30th seconds following the current second, to better fit the delay characteristics of

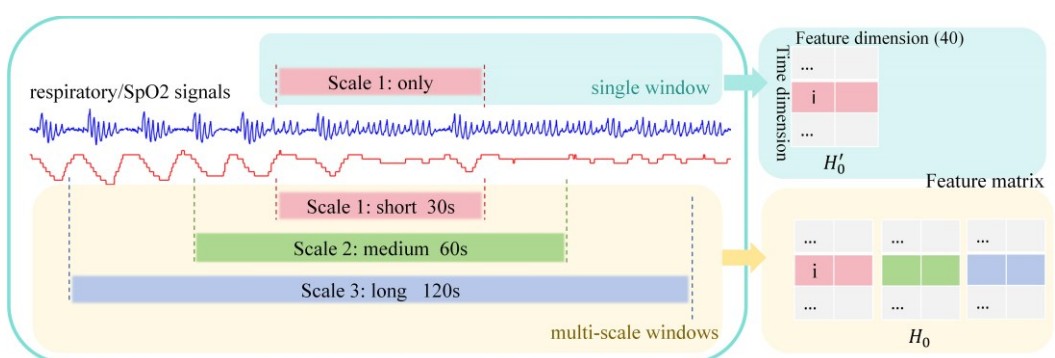

Fig. 3. Multi-scale feature extraction strategy versus single scale feature extraction.

SpO2 signals. The $\mathbf{H}_2 \in \mathbb{R}^{N \times 200}$ contains all the features of SpO2 signals, as shown in Fig. 4c. Finally, $\mathbf{H}_1$ and $\mathbf{H}_2$ are concatenated along the time dimension and the combined feature matrix $\mathbf{H}_{12} \in \mathbb{R}^{N \times 400}$ are obtained.

### D. Feature interaction with XGBoost

*1) Data balance and feature normalization:* We applied random oversampling to the apnea samples in the training dataset to balance the apnea and non-apnea sample ratio [28].In addition, we applied min-max normalization to each feature column of $\mathbf{H}_{12}$ to scale the elements in $\mathbf{H}_{12}$ to range [0,1], helping the model converge faster [29].

*2) Feature interaction with XGBoost*

XGBoost is an efficient gradient boosting decision tree (GBDT) algorithm [30]. Previous studies show that XGBoost performs well in tabular data processing without elaborate tuning [31]. The objective function of XGBoost $L(\boldsymbol{\theta})$ can be written as:

$$L(\boldsymbol{\theta}) = \sum_{i=1}^{M} \ell(y_i, \hat{y}_i) + \Omega(f) \qquad (10)$$

where, $\ell$ represents the logistic loss, $y_i$ is the true value, $\hat{y}_i$ is the predicted value, $M$ refers to the total number of samples, and $\Omega(f)$ is the regularization term. $\Omega(f)$ is designed to control the complexity of the trees to prevent overfitting:

$$\Omega(f) = \gamma T + \frac{1}{2}\lambda \sum_{j=1}^{T} \omega_j^2 \qquad (11)$$

where $T$ is the number of leaf nodes, $\omega_j$ is the weight of the leaf node, $\gamma$ and $\lambda$ are regularization hyperparameters. Through feature splitting process, the XGBoost captures interaction embedded in these features with tree-based structure [32]. XGBoost employs iterative learning between trees, which allows each tree to interact with higher-order features learned by the previous tree, enabling more complex feature interactions and fusion.

## III. RESULTS

When dealing with unbalanced samples, accuracy (Acc) alone cannot accurately evaluate the performance of a model. In this study, we utilize recall (Rec), precision (Pre), and the macro comprehensive metric F1-score to comprehensively measure the diagnostic performance.

### A. Ablation study

To validate the effectiveness of our proposed strategy, the ablation experiments are conducted on our collected dataset. We employ four-fold cross-validation to assess the effectiveness of the proposed Ms&Ne. The data are grouped at the subject level to ensure that samples from the same patient will not appear in the training and testing datasets simultaneously, thereby avoiding information leakage [34].

TABLE III indicates that at the second-by-second level, the simple fusion of respiratory signal and SpO2 can improve the F1-score 0.1188 ,compared with either single signal. Moreover, the proposed Ms&Ne strategy increase the F1-score by another 0.0730 to 0.7713, acquiring a more reliable result. Furthermore, both the Ms and Ne strategies demonstrate significant enhancements in fusion performance, while their combined application achieves optimal results.

We visualize the detection results to elucidate the specific advantages of Ms&Ne, as shown in Fig. 5.The event marked in box 1 in Fig. 5a2 shows that Ms&Ne can recall the missed events with the help of multi-scale information. These advantages are also evident in the area marked in box 2 in Fig. 5b2, where Ms&Ne eliminated the false positive.

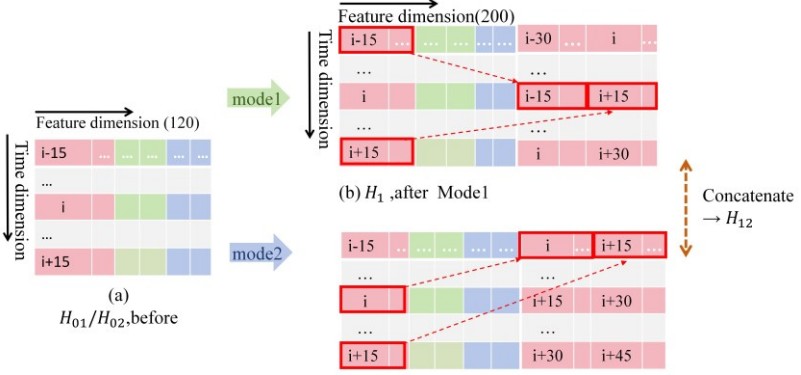

Fig. 4. Neighboring short-term feature enhancement (where mode1 is designed for radar and, mode2 for SpO2 signals, respectively)

TABLE III
APNEA EVENT DETECTION OF 4-FOLD CROSS VALIDATION AT SECOND-BY-SECOND LEVEL (MEAN ± STANDARD DEVIATION)

|  |  | Acc | Pre | Rec | F1-score |
|---|---|---|---|---|---|
| Ms&Ne | Radar+SpO2 | **0.9302**(±0.0084) | **0.8038**(±0.0304) | **0.7416**(±0.0156) | **0.7713**(±0.0217) |
| Ms | Radar+SpO2 | 0.9258(±0.0077) | 0.7905(±0.0339) | 0.7237(±0.0220) | 0.7555(±0.0272) |
| Ne | Radar+SpO2 | 0.9282(±0.0082) | 0.8025(±0.0309) | 0.7275(±0.0158) | 0.7631(±0.0222) |
| Without Ms or Ne | Radar+SpO2 | 0.9109(±0.0088) | 0.7539(±0.0442) | 0.6503(±0.0272) | 0.6983(±0.0346) |
|  | SpO2 | 0.8727(±0.0135) | 0.6098(±0.0543) | 0.5528(±0.0322) | 0.5795(±0.0410) |
|  | Radar | 0.8765(±0.0154) | 0.6580(±0.0623) | 0.4700(±0.0274) | 0.5480(±0.0374) |

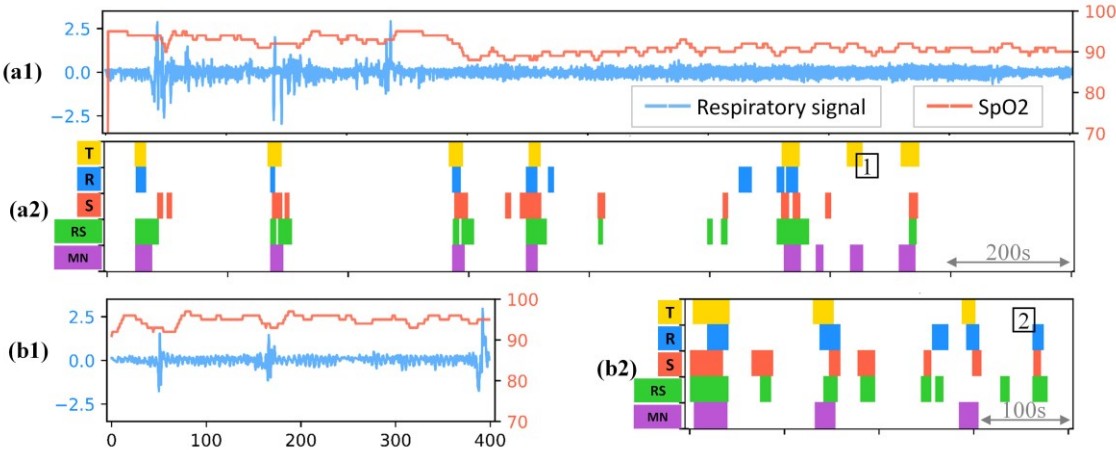

Fig. 5. Predicted apnea distribution with corresponding respiratory and SpO2 signals. (a1) Respiratory and SpO2 signals of one patient, (a2) predicted results for a1, the same below, (b1) zoomed-in view of respiratory signal and SpO2 at 100s scale, (b2) predicted results for b1(Colored areas indicate the presence of apnea events, otherwise, the state is considered normal. T→True label, R→Radar without Ms&Ne, S→SpO2 without Ms&Ne, RS→Radar+SpO2 without Ms&Ne, MN→Radar+SpO2 with Ms&Ne strategy.

TABLE IV
COMPARISON WITH CURRENT STATE-OF-THE-ART RESEARCH (MEAN ± STANDARD DEVIATION)

| Reference | Method | OSA detection | | |
| --- | --- | --- | --- | --- |
| | | Granularity | F1-score | Kappa |
| Toften et al.[23] | LSTM | **1s** | 0.5762(±0.0366) | 0.5105(±0.0337) |
| Ma et al.[19] | CNN-LSTM | 30s segment (decision fusion) | 0.7099(±0.0052) | 0.6583(±0.0056) |
| | | 30s segment (feature fusion) | 0.7255(±0.0078) | 0.6757(±0.0074) |
| Wang et al. [20] | R-CNN-based | event→**1s** | 0.7601(±0.0102) | 0.7142(±0.0120) |
| Our work | Ms&Ne | **1s** | **0.7713**(±0.0217) | **0.7300**(±0.0191) |

**Note:** The abnormal segments detected by the segment-level method in [19] are inconsistent with the strict definition of apnea events. Here, we convert the event-level results from [20] to the second-by-second level and perform the calculation by merging the multi-class apnea types into a binary classification.

## B. Comparative experiments

We compare our method with state-of-the-art studies for OSA diagnosis using millimeter-wave radar and SpO2 signals on our dataset, and the key results are listed in TABLE IV. The segment-based method divides the original data into equal-length segments and classify each segment into apnea and non-apnea (or more apnea types), while the 1s-based method classifies apnea and non-apnea (or more apnea types) at each second. The latter retains the integrity of apnea events and has higher granularity.

In [23], data collection was performed using the Home Sleep Apnea Test (HSAT) [35] method, from 14 individuals, which is a simplified version of PSG testing. It appears to show some incompatibility when using more standard and complicated PSG recordings, achieving an F1-score of only 0.5762 for the second-by-second results. In [19], radar and SpO2 were fused at both the feature level and decision level, and the feature-level fusion method achieved a better F1-score of 0.7255 on our dataset. However, the results are acquired at 30s segment-level, which offers lower granularity, leading to the truncation of complete apnea events. The results in [20] are event-level results, which is efficient for apnea hypopnea index (AHI) calculation. We converted the event-level results into second-by-second results for a more fine-grained comparison. The results indicate that our work achieved the highest F1-score of 0.7713 and a kappa of 0.7300.

## IV. CONCLUSION

This study proposes Ms&Ne, i.e., the multi-scale feature extraction and neighboring short-term feature enhancement strategy designed for OSA detection using millimeter-wave radar and SpO2 signals. Ms&Ne overcomes the limitations associated with limited receptive fields and misalignment between the key features of millimeter-wave radar signals and SpO2 signals, and effectively mitigates the misdiagnosis and missed diagnosis. Experimental results demonstrate that Ms&Ne strategy achieves promising results in apnea event detection, providing substantial support for clinical OSA screening.

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
