# OpenReview forum: "A Lightweight Obstructive Sleep Apnea Detection Method with Millimeter-Wave Radar and Oximeter"
_IEEE.org/EMBS/BHI/2025/Conference — BHI 2025_

### Official Review · Reviewer_C73i · 2025-07-03
**A Lightweight Obstructive Sleep Apnea Detection Method with Millimeter-wave Radar and Oximeter**

**Confidence:** 4
**Clarity Of Writing:** good
**Clinical Significance:** good
**Methodological Novelty:** good
**Overall Rating:** 7

**Experiments And Results:**

good

**Questions For The Authors:**

While the paper highlights the benefits of the Ms&Ne strategy, sensor fusion for OSA detection is not entirely novel. Prior work also explored the value of combining radar and SpO2. What specific aspects of your Ms&Ne strategy differentiate it significantly from existing sensor fusion techniques in the context of OSA detection, and why are these differences critical for improving performance?

You briefly describe the criteria based by AASM v2.6 but how did that help you get your results?

**Strengths:**

Clear Objective and Design: The paper clearly states its goal of developing a user-friendly and fine-grained method for detecting OSA events using millimeter-wave radar and oximeter.
Justification for Approach: The paper effectively justifies the need for alternative OSA detection methods due to the limitations of polysomnography (PSG).
Data Collection and Ethical Considerations: The study clearly describes the data collection process and mentions ethical approvals, ensuring responsible research practices.
Feature Engineering and Extraction:
The methodology for extracting respiratory signals from radar data is well-defined.
The use of tsfresh for feature extraction from time-series data is a strength, providing a comprehensive set of features.
The multi-scale feature extraction strategy and neighboring short-term feature enhancement (Ms&Ne) are clearly described and justified.
Appropriate Machine Learning Techniques: The use of XGBoost is well-justified, given its ability to capture complex feature interactions.
Data Balancing and Normalization: Addressing data imbalance with random oversampling and using min-max normalization are standard practices that enhance the model's performance.

**Summary Of The Paper:**

This paper presents a method for detecting obstructive sleep apnea (OSA) using millimeter-wave radar and oximeter data. The method introduces a multi-scale feature extraction strategy and neighboring short-term feature enhancement (Ms&Ne) to fuse data from the two sensors, addressing signal misalignment due to oxygen desaturation delay. Key features are extracted at long, medium, and short scales. The eXtreme Gradient Boosting (XGBoost) algorithm is used for tree-based feature interactions. Clinical trials involving 121 patients demonstrate that the method achieves an F1-score of 0.7719 for OSA detection at the second-by-second level. The Ms&Ne overcomes the limitations associated with limited receptive fields and misalignment between the key features of radar signals and SpO2 signals, and effectively mitigates the misdiagnosis and missed diagnosis.

**Weaknesses:**

-Limited Scope of Comparison: You compare it with studies that use different type of metrics or use a different method with your same metric that makes it hard to compare the two types of data. You should at least find the same data-set used with different methods to prove yours performs better with the same set.

-Limited Novelty: The single usage of a sensor combined with the new method might be a good innovation, combining two different sensors in one signal. The sensor fusion has been used, so its utility is not new.

---

### Official Review · Reviewer_vTWV · 2025-07-13
**Review of A Lightweight Obstructive Sleep Apnea Detection Method with Millimeter-wave Radar and Oximeter**

**Confidence:** 4
**Clarity Of Writing:** good
**Clinical Significance:** great
**Methodological Novelty:** great
**Overall Rating:** 7
**Final Rating:** 7

**Experiments And Results:**

great

**Questions For The Authors:**

On clinicaltrials.gov the recruitment of this study is still ongoing and the target N is 150. If any recordings or patients were omitted from the dataset used in this paper for any reason, please ensure that those reasons are outlined in the manuscript.
Is there a biological, or other explanation for the timescales chosen for short, mid, and long term feature receptive fields?
Oversampling to account for data imbalance is mentioned, but you don't give a clear description of the labels that are available (second-by-second labels, what do they represent, what is the class ratio)?
Was there a reason for selecting a balanced set of 40 and 40 features from each data source? This question is just my curiosity.
In my experience, text columns usually span from the top of the page to the bottom before moving to the next column position. Before Fig 3. your text moves from the left column to the right column despite the cursor not being at the end of the page. The same thing happens before table IV. Can you please review the formatting guidelines?
Please review the text in the introduction, there are some small grammatical errors. e.g. FMCW acronym not defined, "...while protecting the privacy." - maybe protecting the patient's privacy? There are some others too, so please review carefully.
In figure 5, the yellow light shade indicating that the subject is awake, is not visible or is missing? Please review.

**Strengths:**

This is a very interesting study and the results show a lot of promise. The size of the dataset utilized validates the use of this multi-modal technique, considering previous works used considerably smaller datasets.
The paper is well structured, and the figures support the explanations and results. The ablation study and supporting figure 5 really clearly highlights the impact of the novel algorithm in this work.

**Summary Of The Paper:**

This paper details a method of identifying sleep apnea with a radar to sense chest movements and an SPO2 monitor to identify more subtle apneas, this is a further validation of an experimental setup from reference [23] in the manuscript. The experiment utilizes data from a study of 121 patients and compares with simultaneous PSG recordings as ground truth. The proposed algorithm extracts short and long term features from the multi-modal data.

The novelty is in the design of an algorithm that incorporates both short and long term features, and in their validation of this method (combining radar and SPO2 recordings) instead of traditional PSG using a large dataset.

**Weaknesses:**

Some additional information on the dataset utilized in this study would support the work further. See my question regarding any omission of patients. How long were they recorded for? Was there any pruning or cleaning done to the recorded data, was the data annotated by one person or multiple people. Did all patients experience apnea events during the recordings?

---

### Official Review · Reviewer_KA1m · 2025-07-18
**A Lightweight Obstructive Sleep Apnea Detection Method with Millimeter-wave Radar and Oximeter**

**Confidence:** 4
**Clarity Of Writing:** fair
**Clinical Significance:** good
**Methodological Novelty:** good
**Overall Rating:** 6

**Experiments And Results:**

good

**Questions For The Authors:**

1. Did the rule-based postprocessing method improve the metrics? Did you also apply that rule-based postprocessing method to
the other models when comparing against them in Table IV?

2. Did you consider any other ML model besides XGBoost? Such as CatBoost or LightGBM?

**Strengths:**

1. The work is novel by deviating from the paradigm that only Polsomonography is effective at identifying OSA.

2. The work proposes a new method Ms&Ne which leverages multi-scale feature extraction and neighboring short-term feature enhancement.
The justification for using these two in Ms&Ne were motivated by existing knowledge such as guidelines by AASM. Then, experimental results
from an ablation study performing cross-validation emperically demonstrated that Ms&NE performed better than without.

3. The work applies existing research/models on their custom collected dataset and they demonstrate that Ms&NE outperforms those models
on their dataset.

4. The authors further justify the effectiveness of the Ms&Ne method in Figure 5, which provides explainability into why
Ms&Ne performs better than other methods.

**Summary Of The Paper:**

This work presents an alternative method for detecting sleep apnea by using millimeter-wave radar and oximetry data. The authors propose a novel method of fusing the two modalities together named Ms&Ne by leveraging known discoveries. Then, a comprehensive
list of time series features from both modalities are fed into an XGBoost model. The dataset used was collected specifically for the work and
the data acquisition details are illustrated in the work. The authors demonstrate that their new technique Ms&Ne combined with XGboost outperforms
existing models aimed at tackling OSA detection.

**Weaknesses:**

1. Table IV should also have mean/standard deviation bars like Table 3 (perhaps you need to perform cross-validation to
make Table IV)

2. Few nitpicky details like grammar/typos (for example "The clinical data were collected at Shanghai Sixth
People’s Hospital Affiliated to Shanghai Jiao Tong University
School of Medicine. And 121 records are valid in total for analysis" should be "The clinical data were collected at Shanghai Sixth
People’s Hospital Affiliated to Shanghai Jiao Tong University
School of Medicine and 121 records are valid in total for analysis") and also the figures are low resolution and could be improved
by vectorizing the images into pdf format.

---

### Official Review · Reviewer_2Fo1 · 2025-07-18
**fine-grained obstructive sleep apnea detection**

**Confidence:** 4
**Clarity Of Writing:** great
**Clinical Significance:** great
**Methodological Novelty:** good
**Overall Rating:** 7

**Experiments And Results:**

good

**Questions For The Authors:**

Paper is missing some details that I would like to know more if possible.

1.The paper does not report patient demographics like age, sex, or OSA severity.
2. It lacks details on the number of apnea versus non-apnea samples.
3. Event-level clinical metrics like AHI accuracy are not fully presented.
4. There is no external validation or independent test set evaluation.
5. The paper does not separate the individual impact of multi-scale and neighboring feature strategies.

**Strengths:**

Paper includes the use of a real-world clinical dataset with fine-grained, second-by-second apnea detection, offering higher resolution than prior segment-based methods. Its combination of multi-scale feature extraction and neighboring enhancement with a lightweight XGBoost model leads to clear performance improvements while maintaining computational efficiency.

**Summary Of The Paper:**

Paper does not introduce a fundamentally unique idea as such. It builds on established concepts of combining millimeter-wave radar and oximeter signals for OSA detection. However, paper certainly contributes a practical methodological refinement through the multi-scale and neighboring enhancement (Ms&Ne) strategy, improving fine-grained, second-level detection accuracy and addressing known issues like SpO₂ signal delay.

**Weaknesses:**

Nits:
1. The method is described as “user-friendly”, but it still requires radar equipment and contact-based SpO₂ sensors, limiting practical ease of use.
2. The paper claims to “address signal misalignment”, yet provides no direct measurement of alignment improvement, only indirect performance gains.
3. It refers to “clinical trials with 121 patients”, whereas the study is actually a retrospective data analysis, not a formal clinical trial.
4. The claim of “substantial support for clinical screening” is overstated since the method is tested only in a single controlled hospital setting without real-world validation.